# Enhancing Dielectric Properties, Thermal Conductivity, and Mechanical Properties of Poly(lactic acid)–Thermoplastic Polyurethane Blend Composites by Using a SiC–BaTiO_3_ Hybrid Filler

**DOI:** 10.3390/polym15183735

**Published:** 2023-09-12

**Authors:** Eyob Wondu, Geunhyeong Lee, Jooheon Kim

**Affiliations:** 1Department of Intelligent Energy and Industry, Chung-Ang University, Seoul 06974, Republic of Korea; wendueyoba@gmail.com; 2School of Chemical Engineering and Material Science, Chung-Ang University, Seoul 06974, Republic of Korea; rmsgud96@naver.com; 3Department of Advanced Materials Engineering, Chung-Ang University, Anseong-si 17546, Republic of Korea

**Keywords:** thermoplastic polyurethane, polylactic acid, thermal conductivity, dielectric constant

## Abstract

A composite of polymer blends—thermoplastic polyurethane (TPU) and poly(lactic acid) (PLA)—and BaTiO_3_–SiC was fabricated. BaTiO_3_ particles were used to improve the dielectric properties of the composite materials, whereas SiC was used to enhance thermal conductivity without altering the dielectric properties; notably, SiC has a good dielectric constant. The surfaces of the filler particles, BaTiO_3_ and SiC particles, were activated; BaTiO_3_ was treated with methylene diphenyl diisocyanate (MDI) and SiC’s surface was subjected to calcination and acid treatment, and hybrid fillers were prepared via solution mixing. The surface modifications were verified using Fourier transform infrared spectroscopy (the appearance of OH showed acid treatment of SiC, and the presence of NH, CH_2_, and OH groups indicated the functionalization of BaTiO_3_ particles). After the extruded products were cooled and dried, the specimens were fabricated using minimolding. The thermal stability of the final composites showed improvement. The dielectric constant improved relative to the main matrix at constant and variable frequencies, being about fivefold for 40% BaTiO_3_–SiC–TPU–PLA composites. Upon inclusion of 40 wt.% MDI functionalized BaTiO_3_–SiC particles, an improvement of 232% in thermal conductivity was attained, in comparison to neat TPU–PLA blends.

## 1. Introduction

The improvement of existing power and electrical systems requires excellent energy storage devices, and researchers are seeking compact materials with high heat dissipation capability for use in the fabrication of energy storage devices. Electrostatic capacitors, which are used in large defibrillators, power supplies, pulse networking, actuators, and electric vehicle inverters, are known to have a fast charge–discharge rate, low dielectric losses, and high electric fields [1,2,3,4,5,6,7,8,9,10,11,12,13]. In particular, for use in electronic packaging, actuators, and capacitors, researchers focus on developing materials with high dielectric constants [1,4,5]. Notably, despite technological advancements in the field of electronics, thermal management of materials remains poor, and, hence, excess heat generated in electronic devices is not effectively dissipated into the environment. The miniaturization of electronic devices has made it imperative to develop ways to effectively dissipate heat [7,14,15,16,17,18,19,20,21,22,23,24,25,26,27,28]. Barium titanate possesses a higher dielectric constant than other types of ceramic fillers. Nevertheless, because of its inherent low thermal conductivity, it is incapable of dissipating the heat generated when it is used in miniaturized devices. To enhance the thermal conductivity of materials, researchers are using various types of fillers, including ceramic fillers [15,25,29,30,31,32,33,34], metallic fillers [9,35,36], carbon materials [37,38,39,40], and mixtures of these fillers [19,21,24,26,28,41,42].

Moreover, high dielectric energy storage systems show excellent performance when used as capacitors because of their fast charge–discharge rate, high power density, and long lifetime [43,44,45]. The inherent low thermal conductivity and dielectric constant of polymer materials can be augmented by incorporating filler materials in the polymer matrix. However, usually, fillers with a high thermal conductivity mostly hold a lower dielectric constant, and fillers having high dielectric constants possess a lower thermal conductivity. For instance, BaTiO_3_ possesses a high dielectric constant, exceeding 5000, but has a low thermal conductivity (<7 W/(m.K)) [5], and CNTs have a thermal conductivity above 3000 W (m.K) but a dielectric constant below five (in a range similar to that of a neat polymer matrix). Khadim et al. demonstrated nanostructures of SiO_2_ and SrTiO_3_ doped on polystyrene polymer materials to enhance dielectric properties and optical characteristics and obtained improved optical properties and dielectric constants [46].

Wu et al. demonstrated a dielectric constant improvement using immobilized graphene oxide on the surface of hexagonal boron nitride and epoxy matrix [1]. The immobilized graphene oxide was fixed on the surface of hexagonal boron nitride which yielded a higher dielectric constant as compared to the neat epoxy matrix. Lin et al., used silicon carbide filler particles treated with polyurea/3-aminopropyltriethoxysilane and polyetherimide to obtain improved dielectric constant composites [3]. Due to the interfacial adhesion improvement of silicon carbide particles by the surface treatments made using polyurea/3-aminopropyltriethoxysilane, the dielectric constant of the composites of polyetherimide and surface-treated SiC improved significantly. 

Several researchers have investigated the parallel improvement of thermal conductivity and the dielectric constant. For instance, Zhou et al., investigated the improvement of the dielectric constant and thermal conductivity by utilizing structured core double-shell fillers [6]. They encapsulated aluminum metal with amorphous aluminum oxide and found that the created double shell provided an improvement in the dielectric constant and thermal conductivity. Composites of epoxy and hybrid fillers of silicon carbide coated with silicon dioxide and boron nitride composites were investigated by Zhao et al. [13]. Boron nitride was used to improve the thermal conductivity, whereas the dielectric constant was improved by the silicon carbide coated with silicon dioxide and obtained a thermal conductivity of 0.76 W/(m.K) and a dielectric constant of 8.9. Yang et al. provided a simultaneous increment of the dielectric constant and thermal conductivity of nitrile butadiene rubber and boron nitride filler composites [5]. The functionalization of nitride butadiene rubber and boron nitride was achieved through the improvement of the surface of boron nitride with poly(dopamine) through subsequently functionalizing boron nitride surfaces with γ-(2,3-epoxypropoxy) propytrimethoxysilane (KH560). Thermal conductive composites of 0.409 W/(m.K) and higher dielectric constants were obtained. Furthermore, researchers continue to investigate the improvement of both thermal conductivity and the dielectric constant as the ones reported. Recently, Liu et al. constructed high thermal conductivity and dielectric constant composites from boron nitride filler particles and an epoxy resin polymer matrix [11]. Both thermal conductivities and the dielectric constant were improved simultaneously upon encapsulating the surface of boron nitride particles with benzene-1,3,5-triyl tri benzoate. 

While fabricating filler-material-containing polymer composites, the mechanical properties of the composites deteriorate upon increasing the filler particle loading percentage beyond a threshold. In our previous research work [18], we presented that the inclusion of filler materials above the threshold level degraded the mechanical characteristics. 

In the current study, we attempted to incorporate hybrid fillers that could simultaneously improve the thermal conductivity and dielectric properties of polymer materials getting the motivation from the drawbacks of few researchers in simultaneous improvements in dielectric properties and thermal conductivity. Such materials have the potential to be used in electronic packaging industries. Hybrid fillers were prepared by treating the surfaces of two fillers, namely BaTiO_3_ and SiC, to make a reactive substarate through calcination and a solution method. The surface modification of BaTiO_3_ filler particles was considered to examine the functionalization of diisocyanate groups, and the SiC surface modification was checked for the availability of reactive groups, OH, on its surface. TPU–PLA blend composite materials and the BaTiO_3_–SiC hybrid filler were prepared via the process of low-speed melt–extrusion with constant temperature profiles. BaTiO3 was chosen to enhance the dielectric performance and SiC was utilized to improve the thermal conductivity of the composites. 

## 2. Materials and Methods 

### 2.1. Materials 

TPUs, polylactic acid (PLA), and matrices of the produced composites were procured from Dong-Sung Chemical, Ulsan, Republic of Korea. Acetone, sodium hydroxide (NaOH), ethanol, 4,4′-Diphenylmethane diisocyanate (MDI), and hydrochloric acid (HCl) were obtained from Dae-Jung Chemical and Metal Co. Ltd., Seoul, Republic of Korea; Alfa Aesar, MI, USA, provided barium titanate (99%, BaTiO_3_) particles. Furthermore, dibutyltin dilaurate (DBTDL)m, 3-aminopropyltrimethoxysilane (APTMS), and silicon carbide (SiC, 40 µm) were provided by Sigma-Aldrich, St. Louis, MI, USA. 

### 2.2. Methods

Hydroxyl (OH) groups were introduced on the unreactive BaTiO_3_ particles’ surfaces to promote the reaction of the particles with the diisocyanate groups of MDI (Figure 1a). The hydroxyl treatment involved the use of a 4 M NaOH solution. NaOH was dissolved in deionized (DI) water, and BaTiO_3_ particles were added to the solution under stirring at 80 °C. The hydroxyl-group treatment was performed upon stirring at higher speeds for 24 h. Finally, the OH-treated solution was washed, and the BaTiO_3_ particles were isolated through vacuum filtration. The solution was washed with DI water until it became neutral while filtering it. The filter cake was dried in an oven for 24 h at 60 °C. The OH-treated BaTiO_3_ particles were then subjected to MDI treatment to make the OH groups react with the diisocyanate groups of MDI. Acetone was chosen as the solvent since MDI dissolves in it when DBTDL is used as a catalyst. The reaction took place at 60 °C for 6 h. Following the MDI group’s attachment to the BaTiO_3_ particles, separation of the solution was done from the solvent by a vacuum-filtration technique, and drying of the filter cake followed in an oven for 24 h at 60 °C. The treatment of the filler particles was completed upon treating the SiC particles via calcination at 400 °C once the surface was dissolved in DI water for cleaning. 

The calcinated DI-treated SiC particles were then subjected to acid treatment to attach hydronium groups on their surface (obtained from the hydrolysis of HCl), to promote the reaction of the hydronium groups with the remaining OH groups during the fabrication of the hybrid filler (Figure 1b). The hybrid filler comprising MDI-treated BaTiO_3_ particles and calcinated hydronium-modified SiC particles was prepared using ethanol as a solvent in an oil bath at 65 °C, and it was dried at 60 °C after separating the solvent from the particles through vacuum filtration. An amount of 60 wt.% of BaTiO_3_ particles was used to fabricate the hybrid fillers, with SiC accounting for the remaining weight percentage. 

The composites were produced using the melt–extrusion method at a low speed to decrease the viscosity. The scheme for the fabrication of the composite is displayed in Figure 2. Blends were fabricated at various ratios of TPU and the PLA polymer matrix, and the fillers were added to the composite ratio having a negotiable mechanical property (determined after analyzing the elongation at break and tensile strength). The filler-particle ratio is fixed at 60 wt.% BaTiO_3_ and 40 wt.% SiC, whereas the composites were varied in a step size of 10 wt.% and the blends were varied from 30–70 TPU and PLA in weight percentage. Upon observing a drop in tensile strength, the TPU–PLA blend used to fabricate the composite is fixed at 60 wt.% TPU and 40 wt.% PLA. After samples were prepared using the melt–extrusion method, specimens of a dog-bone-shape were obtained with a minimolding machine, and the shaped dog-bone-shaped specimens were subjected to 1 kN loading force in the UTM machine to ascertain their tensile properties at a loading rate of 5 mm/s.

## 3. Characterization

The effectiveness of modification of the surface SiC particles using HCl and the attachment of MDI to functional groups on the surface of BaTiO_3_ particles were examined using Fourier transform infrared spectroscopy (FTIR; Nicolet iS5, Thermo Fisher Scientific, Seoul, Republic of Korea); the attenuated total reflectance (ATR) method was employed. Field-emission scanning electron microscopy (FE-SEM; Sigma, Carl Zeiss, Oberkochen, Germany) was employed to analyze the morphology of the MDI groups attached to the BaTiO_3_ particle’s surface, and the elemental composition of the MDI groups was determined using X-ray photoelectron spectroscopy (XPS; K-Alpha, Thermo Fisher Scientific). The amount of MDI groups attached to the surfaces of the BaTiO_3_ particles was determined through thermogravimetric analysis (TGA; Thermogravimetric Analyzer 2050, TA Instruments, New Castle, DE, USA). 

Thermal degradation properties of neat polymer blends and SiC–MDI-treated BaTiO_3_ composites were investigated through TGA. Thermal degradations were recorded in the 25–600 °C range of temperatures by utilizing nitrogen gas to keep the environment inert. Morphological FE-SEM investigations were performed for the composites after fracturing the specimens by using liquid nitrogen. The tensile strength and mechanical properties of the prepared composite materials were explored by a universal testing machine (UTM; model UTM-301, R&B Corp., Daejeon, Republic of Korea) at a crosshead speed of 5 mm/min and a loading force of 100 MPa, following the ASTM D5262 standards [42]. 

Dynamic mechanical analysis (DMA; Dynamic Mechanical Analyzer 8000, PerkinElmer, Shelton, CT, USA) was performed to investigate the viscoelastic nature of the composites in a temperature range of −80 to 150 °C (3 °C/min heating), and an impedance analyzer (4294A, Agilent Zurich, Switzerland) was employed to investigate the dielectric constant of the composites at room temperature. The composites’ thermal conductivity was determined through laser flash analysis (LFA 467 Hyper-Flash, Netzsch Instrument Co., Selb, Germany) at ambient temperature. 

## 4. Results and Discussions

The FTIR spectra of the SiC particle surface treatment using HCl are provided in Figure 3a. The peak at a wavenumber of 1080 cm^−1^ and 1260 cm^−1^ is ascribed to functional groups of C-O and OH. The effect of MDI functionalization on the surfaces of BaTiO_3_ particles was studied using FTIR spectroscopy, FE-SEM, TGA, and XPS. Figure 3 demonstrates the FTIR and XPS spectra of MDI-treated BaTiO_3_ particles. The wavenumber range of 400 to 4000 cm^−1^ was studied to verify the functional groups of MDI, which were utilized for the treatment of the surfaces of BaTiO_3_ particles and were apparent in the spectra (Figure 3b). The peak at 3470 cm^−1^ was ascribed to the amine groups (NH groups) obtained from the diisocyanate groups. The isocyanate functional groups observed at 2260 cm^−1^ originated from the MDI groups that are utilized to functionalize the surfaces of BaTiO_3_ particles. The 2960 cm^−1^_,_ and 3310 cm^−1^ wavenumbers are attributed to methylene groups (CH_2_) and OH groups, which are characteristic peaks arising from the NaOH. The emergence of both amine groups and diisocyanate groups confirmed the functionalization of the surfaces of BaTiO_3_ particles using MDI. 

Additionally, XPS analysis is utilized to distinguish between pristine BaTiO_3_ particles and MDI-functionalized BaTiO_3_ particles. The particles are dried before XPS examination to thwart moisture accumulation. Figure 3c,d demonstrates the outcomes of the XPS examination. Wider survey scans of the XPS examination of the pristine and MDI-functionalized BaTiO_3_ particles are displayed in Figure 3c. 

The 402 eV binding energy was ascribed to N1s arising from the diisocyanate group of MDI groups introduced during the surface functionalization of BaTiO_3_ particles. A decrease in the O1s intensity at the binding energy of 531 shows the precise functionalization of BaTiO_3_ particles with MDI; the oxygen in the MDI groups dissociated while the reaction was taking place, which decreased the intensity of the oxygen of BaTiO_3_. An increase in the intensity of C1s peaks is due to the functionalization of diisocyanate groups that have a higher amount of carbon atoms. The XPS spectra were deconvoluted using the Gaussian fitting method to examine whether they showed the surface modification of BaTiO_3_ particles with MDI groups. Figure 3d shows the deconvolution results for MDI-modified BaTiO_3_ particles. N1s was deconvoluted in the range of 409 and 393 eV binding energies and the results of deconvolution showed a dominant sharp peak conforming to the amine groups at 399 eV (C=N-C). Moreover, the XPS investigation outputs stipulating the atomic percentage are shown in Table 1. As described in the table, the percentage of carbon elements increased for the MDI functionalized BaTiO_3_ particles, as compared to the pristine BaTiO_3_ particles; this might be because of the carbon obtained from the diisocyanate groups. This is supported by the survey peak. Importantly, the atomic percentage of the nitrogen element obtained from the MDI groups is found to be 9.85% in the MDI functionalized BaTiO_3_ particles, whereas that of the pristine BaTiO_3_ particles’ nitrogen atom content is negligible, and the atomic percentage of barium, titanium, and oxygen has decreased upon functionalizing the surface of BaTiO_3_ particles using MDI. 

Furthermore, to determine the amount of MDI groups that are used to functionalize the BaTiO_3_ particles, TGA evaluation was performed in a range of temperatures of 25–600 °C. As shown in Figure 4a, it was found that pristine BaTiO_3_ particles were heat resistant until 600 °C (thermally stable). However, the MDI-functionalized BaTiO_3_ particles specimen degraded at this temperature (9.1 wt.% degradations are observed), which indicated diisocyanate groups accounted for about 9.1 wt.% of the MDI-functionalized BaTiO_3_ particles. The MDI groups started degrading at 222 °C, and this degradation continued linearly till the temperature of 290 °C was attained. Following that, a stable region with an almost zero slope throughout stops the degradation, exiting the residue as the BaTiO_3_ particles reach 320 °C. 

The morphology of the MDI group’s functionalization of the BaTiO_3_ particles’ surfaces was inspected by FE-SEM examination, and it was compared with that of the morphology of pristine BaTiO_3_ particles. The FE-SEM exploration results are illustrated in Figure 4b,c. The morphology of the MDI-functionalized BaTiO_3_ particles’ surfaces is illustrated in Figure 4c, and the surfaces of pristine BaTiO_3_ particles are shown in Figure 4b for facilitating a comparison. It is noteworthy that the surface-functionalized BaTiO_3_ particles were considerably distinctive from the pristine BaTiO_3_ particles, as the MDI-functionalized BaTiO_3_ particles were shielded with MDI molecules (the MDI groups were distributed randomly on the surfaces of the BaTiO_3_ particles, as the attachment of OH groups to the surface of BaTiO_3_ particles helped the filler particles react with the NCO groups from the MDI). This is supported by the XPS and TGA analysis. 

The trends in thermal stability (thermal degradation) of TPU–PLA with MDI-functionalized BaTiO_3_–SiC composites were studied at variable temperatures by employing thermogravimetry in a range of temperatures of 25–600 °C; the surrounding environment was filled with nitrogen to prevent the further oxidation that could happen. The inclusion of the filler particles (MDI-functionalized BaTiO_3_–SiC particles) into the TPU–PLA matrix did not adversely affect the thermal stability; the inclusion of the filler particles enhanced the composite’s thermal stability, as evident in Figure 5c. Three regions of thermal degradation were observed for neat TPU–PLA blends in the TGA examination. The first region of degradation began at 286 °C; that is the temperature at which the TPU soft segments started to degrade. The degradation continued to occur linearly up to 315 °C. The TPU soft segments completely degraded at 315 °C, and after the degradation of the TPU soft segments, the TPU hard segments started degrading. The hard segment chains of TPU and PLA began their region of degradation at 340 °C, and the degradation region continued linearly till the final residue (1 wt.%) was attained at an offset temperature of 420 °C.

The composites of MDI-functionalized BaTiO_3_–SiC particles with TPU–PLA blend matrix showed four regions of degradation: the first region’s onset temperature was 290 °C, and it was analogous with that of the neat TPU–PLA polymer matrix. At this temperature, the TPU soft segment began degradation mutually with the MDI molecules’ amine functional groups. This region of degradation continued with an infinite slope till the TPU soft segment chain’s complete degradation temperature was attained, namely 310 °C. The degradation stage of the following stage was begun from the offset temperature of the soft segments of TPU. In this stage of degradation, MDI and the remaining moisture on the composite’s surface were degraded, that is, at 340 °C, which is taken as its onset temperature and continued until the temperature of 383 °C was reached, leaving hard segments of the TPU–PLA matrix and the BaTiO_3_–SiC filler particles as the residue. The onset region of degradation of the residual TPU hard segments, PLA, and MDI molecules was ruminated to be 380 °C, and this region of degradation continued in a linear fashion until only the last residue, SiC and BaTiO_3_ particles, remained. The final residue amount depended on the amount of loading percentage of each of the filler particles. Notably, in all the regions of the degradation stages, the MDI molecules, which were used for functionalization, degraded the BaTiO_3_ particles, since MDI itself acted and is considered to be hard segments of the TPU matrix [47,48]. 

FE-SEM inspection was utilized to explore the morphology of the liquified nitrogen-fractured specimens. Before the FE-SEM analysis was conducted, platinum was coated on all the samples during the course of analysis to prevent the accumulation of charges. The FE-SEM examination outputs are described in Figure 5a,b. The neat TPU–PLA fractured specimens display a more or less brittle nature, as evident in Figure 5a, which is supported by the UTM analysis. Thus, neat TPU and PLA appeared to be fractured brittle materials in the FE-SEM evaluation. Fortunately, the TPU–PLA composites with MDI-functionalized BaTiO_3_–SiC particles unveiled good interfacial adhesion properties with the surface of the TPU–PLA blend matrix because MDI acted as a compatibilizer between the surfaces of BaTiO_3_ particles and the polyurethane matrix. Figure 5b stipulates the FE-SEM evaluation result for the TPU–PLA with MDI-functionalized BaTiO_3_–SiC particle composites). In addition, the addition of TPU together with the filler particles has improved the ductility of the composites which is supported by the elongation at the break in Figure 6b.

The MDI functionalized BaTiO_3_–SiC particle composites with TPU–PLA showed altered mechanical properties, especially the strain. The elongation at the break and the strength at the break (tensile strength) were explored using a UTM. Figure 6 stipulates the UTM analysis results. Tensile strength was counted as the stress where the specimens lost their elastic nature and broke, while elongation at the break was counted as the strain where the specimens broke under the loading force applied. In this work, for every sample, both the tensile strength and the elongation at the break were determined using four specimens, and the average of the four was counted as the tensile strength and elongation at the break point of the specimen under investigation. The elongation at the break and the tensile strength of neat 30-70, 70-30, 40-60, 60-40, 10TPU–PLA, 20TPU–PLA, 30TPU–PLA, and 40TPU–PLA each represent the numbers; the first number represents TPU content in percentage, and the second one after the hyphen represents the PLA content in percentage were investigated. The numbers accompanying abbreviations represent the amount of filler loading (MDI-functionalized BaTiO_3_–SiC) in percentage (10, 20, 30, and 40 indicate 10, 20, 30, and 40 wt.% of BaTiO_3_–SiC particles being incorporated in 60 wt.% polyurethane and 40 wt.% poly(lactic acid)). 

The average tensile strength of the neat TPU was around 12 MPa [48], whereas that of the neat PLA was around 46 Mpa [47]. The tensile strength of the blend composites decreased slightly, with an increase in the percentage of MDI-functionalized BaTiO_3_–SiC loadings to the blend matrix in comparison to the neat PLA; however, it increased as compared to neat TPU. From all specimens investigated, 10TPU–PLA attained the lowest strength at the break, probably because it had the lowest filler content in the polymer matrix. Figure 6a depicts the tensile strength results. Compared with the neat TPU and PLA blends, the MDI-functionalized BaTiO_3_–SiC TPU–PLA composites had lower tensile strength. The 40TPU–PLA specimens’ average tensile strength was about 17.5 MPa, superior to neat TPU, which might be because of the inclusion of PLA, which had a higher tensile strength. The higher tensile strength of 40TPU–PLA compared with the 10TPU–PLA composites was probably because of the larger number of BaTiO_3_–SiC particles distributed over the surface of the TPU–PLA polymer matrix in the case of the former; this fact is supported by the morphological improvements between the matrix, TPU–PLA blend, and the filler particles improved by MDI functionalization of BaTiO_3_, as displayed in Figure 5b. However, the elongation at the break of the composites dropped once the PLA polymer was included in the composites, owing to the brittle nature of PLA. The elongation at the break is displayed in Figure 6b. As the loading percentage of BaTiO_3_–SiC particles in the matrix of TPU–PLA increased, the brittle nature decreased, inferring that the composites were more ductile than 60-40TPU–PLA, which might be due to the filler particles incorporation in the TPU–PLA blend matrix. However, the elongation at the break point of the TPU–PLA composites with MDI-functionalized BaTiO_3_–SiC particles showed a superior ductile property, compared with TPU–PLA. The elongation at the break of 30-70 was the lowest among all the samples because of the presence of a large amount of PLA matrix in the polymer blend (which was a highly brittle material). A higher elongation at the break was obtained for 70-30, and it was because of the inclusion of more TPU in the blend, which made the material highly ductile. 

The tan δ, storage modulus, of the fabricated TPU–PLA composites with MDI-functionalized BaTiO_3_–SiC particles were explored with DMA to determine the viscoelastic nature of the considered materials in a range of temperatures of –80 to 150 °C. As a pressurizer, nitrogen gas is utilized to flow liquid nitrogen that was used to attain a temperature of −80 °C. The viscoelastic natures of the TPU–PLA composites with MDI-functionalized BaTiO_3_/SiC particles are shown in Figure 7. The storage modulus represents the stiffness of the materials, and tan δ informs the material’s glass transition temperature. The storage modulus of TPU–PLA and TPU–PLA with MDI-functionalized BaTiO_3_–SiC composites is displayed in Figure 7a. The stiffness increased with the weight percentage of MDI-functionalized BaTiO_3_–SiC particles. In particular, 40TPU–PLA had the highest stiffness, and 70-30 had the lowest stiffness, which implied that the storage modulus (stiffness) decreased as the amount of PLA increased. Composites with a higher stiffness than the neat TPU–PLA were obtained, probably because of the inclusion of a larger amount of filler, which had high stiffness. The enhancement of the composites’ stiffness property could be attributed to the highly stiff inherent property of the filler particles, the larger the amount of filler particles in the composites, the larger the storage modulus. This is described in Figure 7a, which shows the storage modulus of the obtained composites. 

We analyzed tan δ to ascertain the glass transition temperature of the prepared composites. Figure 7b shows the tan δ changes observed while evaluating the specimen’s viscoelastic nature. The temperature at which the tan δ reaches a maximum point is believed to be the glass transition temperature. In this study, two glass transition temperature regions were observed, one corresponding to the TPU and the other to the PLA polymer matrix. The height of the glass transition temperature (tan δ peak) depended on the percentage of PLA and TPU. The composite having more PLA had a large peak for the PLA glass transition temperature (70 °C), as shown, and that with more TPU had a large peak for TPU (−16 °C). For TPU–PLA MDI-functionalized BaTiO_3_–SiC composites, the tan δ peak depended on the amount of filler in the polymer matrix (a larger amount of filler particles resulted in a larger tan δ peak height). Thus, among the TPU–PLA MDI-functionalized BaTiO_3_–SiC composites, 40TPU–PLA had the largest tan δ peak. 

The fabricated composites’ ability to store charge was investigated using the impedance analyzer for all the filler loadings of the composite; measurements were performed on two specimens as a function of a variable frequency of 0–10 MHz, and the output is demonstrated in Figure 8. 

The dielectric property as a function of frequency and at constant frequency is shown in Figure 8. The variable-frequency dielectric property explains the occurrence of a higher dielectric constant at lower frequencies, about 66.5 for the 40TPU–PLA composites. When the frequency was increased, the dielectric constant dropped with an infinite slope till it reached the frequency at which the specimen’s dielectric constant became stable, and it remained constant thereafter. The dielectric constant of the TPU–PLA matrix (60-40) started dropping from 5.8 to its constant dielectric constant gaining frequency, i.e., 200 kHz, at which its value became 4.2. The dielectric constant at lower frequencies gradually increased with the increase in the filler-loading percentage. The dielectric constants of the fabricated specimens were also compared at constant frequencies of 1 MHz, shown in Figure 8b, to examine the trend in filler loading at a constant frequency. It was deduced that increasing the filler particles’ percentage loading that was surface functionalized with MDI increased the dielectric constant at higher frequencies. Interestingly, the specimen containing 40 wt.% MDI-functionalized BaTiO_3_–SiC filler particles showed a dielectric constant at 1 MHz of 20.4, which might have been because of the outstanding dielectric constant derived from the BaTiO_3_ and SiC filler particles and good interfacial adhesion between the TPU–PLA matrix and the filler particles as a result of the inclusion of MDI groups (hard segment of TPU) in the filler particles. 

LFA analysis was employed to examine the thermal conductivity of TPU–PLA MDI-functionalized BaTiO_3_–SiC particles. Polymer materials possess a very low thermal conductivity of less than 0.2 W/(m.K). Generally, thermal conductivity is dependent on the filler orientation with the polymer matrix, defects in the chain of polymers, and the presence of voids. To calculate the thermal conductivity of composite materials, the following equations are applied (Equation (1)): where Ƙ is the thermal conductivity, Cp is the specific heat capacity, δ is diffusivity, and ρ is the density.
Ƙ = ρCpδ(1)

The LFA machine tells the thermal diffusivity depending on the thickness (1.43 mm) and diameter (10 mm), whereas the Cp and ρ are obtained from differential scanning calorimetry and taking the particle as circular disks, respectively. In this study, the effects of incorporation of MDI functionalized BaTiO_3_–SiC particles to TPU–PLA blends on thermal conductivity were investigated. As the loading percentage of MDI-functionalized BaTiO_3_–SiC particles increases, the thermal conductivity increases in a linear fashion. The results are stipulated in Figure 9. Upon inclusion of 40 wt.% MDI-functionalized BaTiO_3_–SiC particles, an improvement of 232% in thermal conductivity was attained in comparison to the neat TPU–PLA blends. This might be due to the inclusion of 40 wt.% SiC in reference to BaTiO_3_ particles of which the total filler is 40 wt.% with respect to the polymer-blend matrix. This amplifies a higher thermal conductivity, from the nature of SiC particles was acquired. Moreover, the improvement of the surfaces of BaTiO_3_ particles with MDI, which acts as a compatibilizer between the polymer matrix and filler particles, has improved the interfacial interaction between the matrix and fillers which, in turn, improved the thermal conductivity. 

## 5. Conclusions

This study focused on the improvement of the dielectric properties and thermal conductivity of TPU–PLA composites having excellent tensile strength and thermal stability using BaTiO_3_ and SiC filler particles. Before fabrication of the composites, the surfaces of filler particles were modified to activate them. Particles of BaTiO_3_ were primarily modified with NaOH to introduce OH groups onto its surface so that these functional groups would react easily with diisocyanate groups (MDI), which would, in turn, react with the soft segment of the TPU matrix. The SiC particles were treated with DI water and acid and then calcined. The treatments were confirmed using FTIR spectroscopy, XPS, FE-SEM, EDS, and TGA, which indicated successful functionalization. The composites were prepared using melt–extrusion methods and subsequent minimolding. The morphological appearance, thermal stability, viscoelastic properties, tensile properties, dielectric properties, and thermal conductivities were investigated. Thermal stability was found to improve upon the inclusion of MDI-functionalized BaTiO_3_–SiC particles. An investigation of mechanical properties showed an improvement in tensile strength compared with neat TPU, but a decrease in this parameter compared with neat PLA. The blend composites possessed a tensile strength in between those of the TPU and PLA. The dielectric constant at 1 MHz showed a fivefold improvement compared with that of the matrix. Upon inclusion of 40 wt.% MDI functionalized BaTiO_3_–SiC particles, an improvement of 232% in thermal conductivity was attained in comparison to the neat TPU–PLA blends.

## Figures and Tables

**Figure 1 polymers-15-03735-f001:**
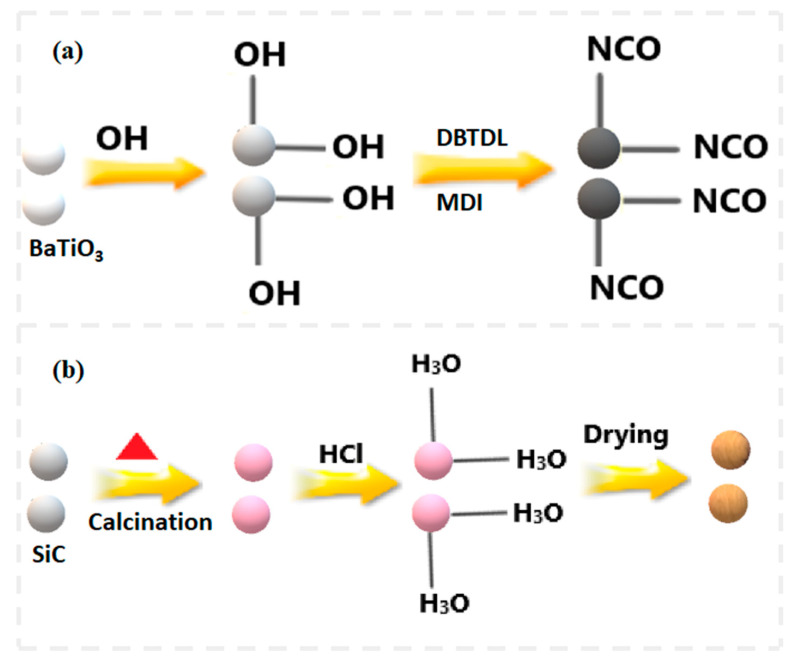
Scheme of surface modification of filler particles: (**a**) Modification of the surfaces of BaTiO_3_ particles with MDI groups; (**b**) Modification of the surfaces of SiC particles with HCl.

**Figure 2 polymers-15-03735-f002:**
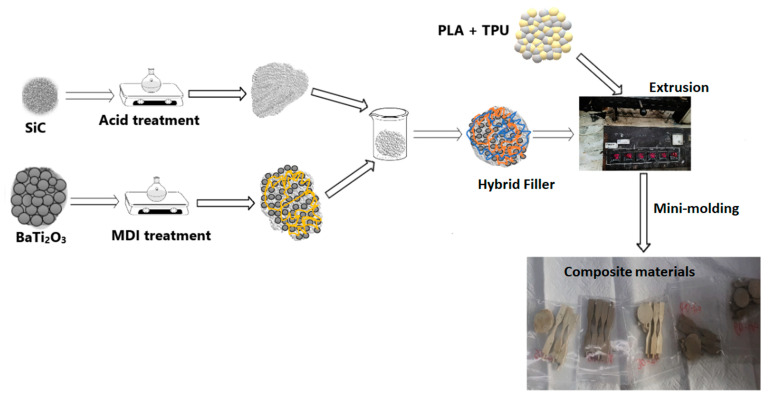
Scheme for fabrication of the composite materials through melt–extrusion methods.

**Figure 3 polymers-15-03735-f003:**
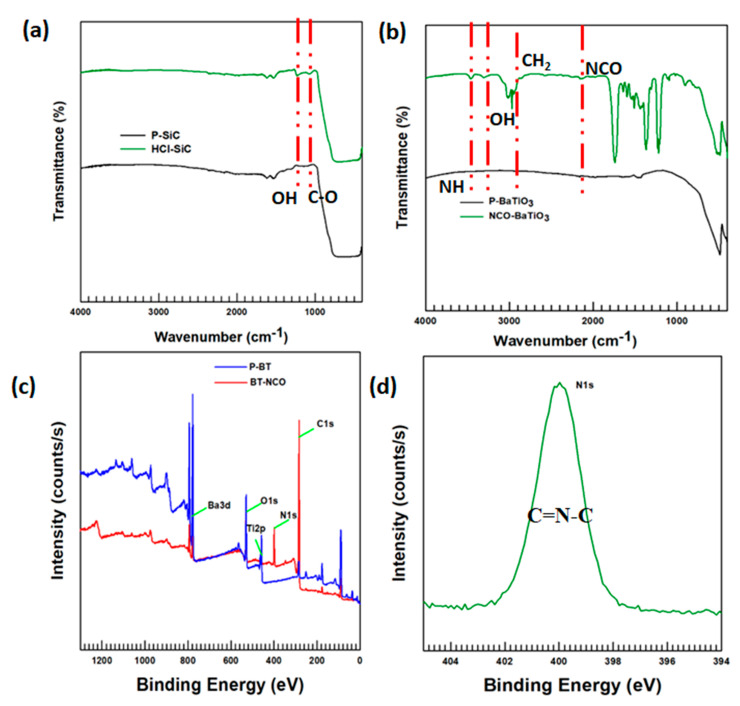
Characterization of filler particles’ surface treatment: (**a**) FTIR investigation of the treatment of surface of SiC with HCl; (**b**) FTIR investigation of the treatment of surface of BaTiO_3_ with MDI; (**c**) XPS investigation survey output; (**d**) Deconvolution of N1s XPS investigation.

**Figure 4 polymers-15-03735-f004:**
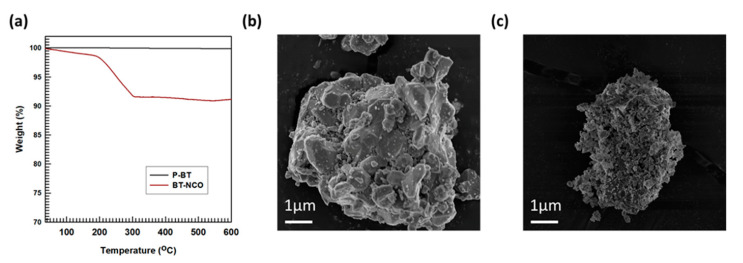
Determination of surface treatment of BaTiO_3_ particles: (**a**) TGA investigation; (**b**,**c**) FESEM investigation of pristine and MDI-functionalized BaTiO_3_ particles, respectively.

**Figure 5 polymers-15-03735-f005:**
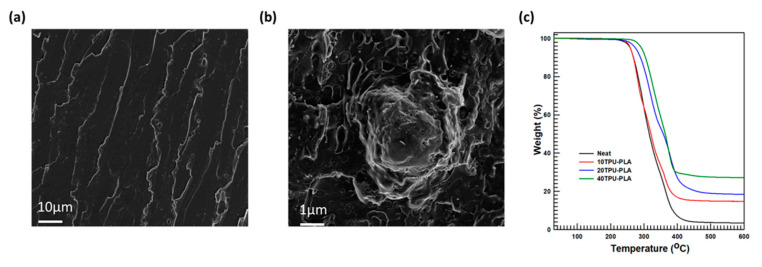
Morphology and thermal stability: (**a**) Neat TPU–PLA blend composite morphology; (**b**) FE-SEM of 40TPU–PLA and (**c**) Thermal degradation study of the composites.

**Figure 6 polymers-15-03735-f006:**
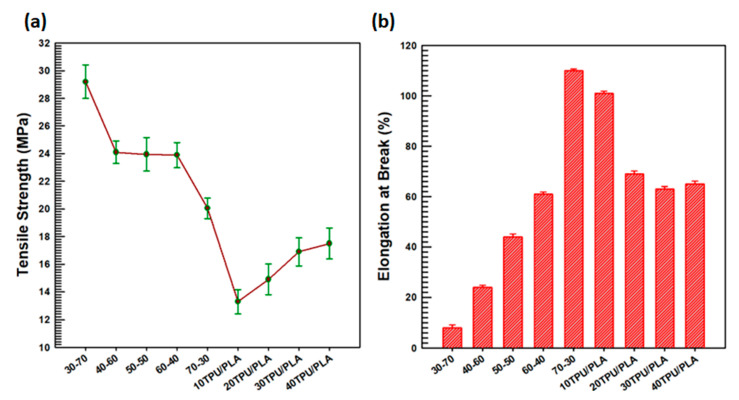
Mechanical properties: (**a**) Tensile strength; (**b**) Elongation at break.

**Figure 7 polymers-15-03735-f007:**
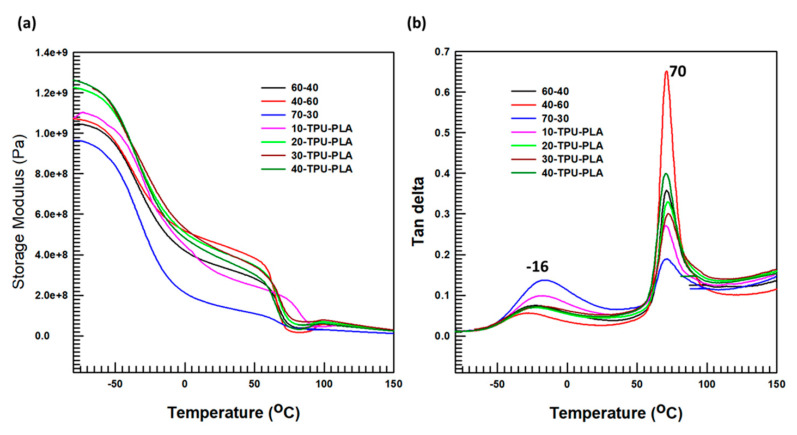
Thermomechanical properties: (**a**) Storage Modulus; (**b**) Tan δ.

**Figure 8 polymers-15-03735-f008:**
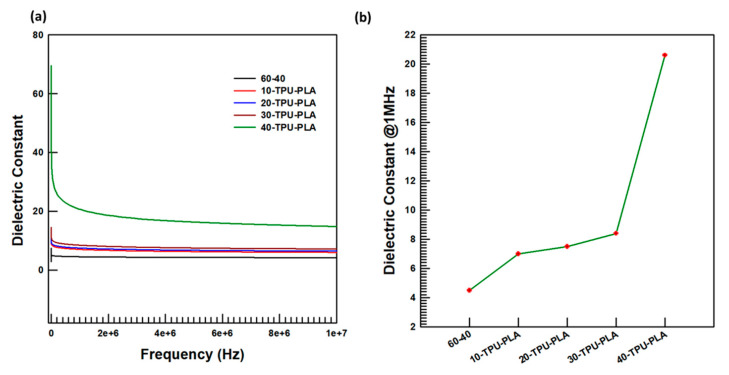
Dielectric constant investigations: (**a**) At variable frequency; (**b**) at a constant frequency, 1 MHz.

**Figure 9 polymers-15-03735-f009:**
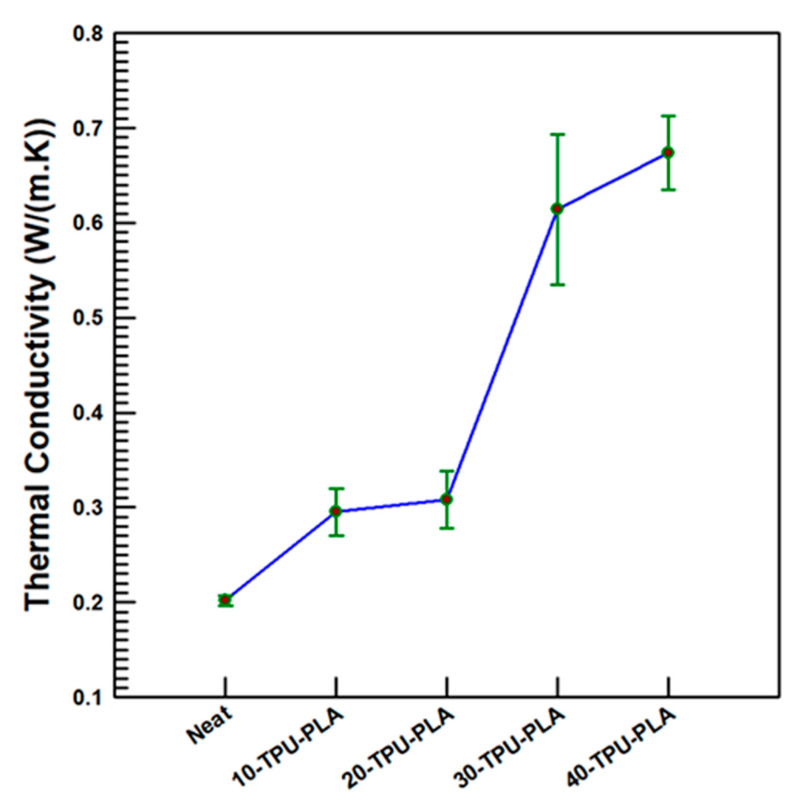
Thermal conductivity of TPU–PLA MDI functionalized BaTiO_3_–SiC composites.

**Table 1 polymers-15-03735-t001:** Elemental atomic percentage and area covered by pristine BaTiO_3_ and MDI-functionalized BaTiO_3_ particles.

Elements	Area Covered (P) CPS.eV (PBaTiO_3_)	Area Covered (P) CPS.eV (MDI-BaTiO_3_)	Atomic % Age (PBaTiO_3_)	Atomic % Age (MDI-BaTiO_3_)
Ba3d	99,627.02	103,710.28	**15.01**	**1.45**
O1s	45,395.88	72,096.52	**45.17**	**12.03**
Ti2p	31,237.5	23,312.42	**12.41**	**1.36**
N1s	–	39,990.67	**–**	**9.85**
C1s	8647.2	180,177.81	**23.72**	**73.96**

## Data Availability

All data used in this study are included in the manuscript.

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
