# Peer review of "Enhancing Dielectric Properties, Thermal Conductivity, and Mechanical Properties of Poly(lactic acid)–Thermoplastic Polyurethane Blend Composites by Using a SiC–BaTiO3 Hybrid Filler"

_polymers, 2023, doi:10.3390/polym15183735_

Round 1

Reviewer 1 Report

Why DOE/Response Surface Methodology was not employed in this study?

Author Response

Dear Reviewer,
Thank you for the insightful comments. 
We have addressed your comments and attached our response.

Sincerely,

Reviewer 2 Report

This is a very good and necessary article, and it can be recommended for publication after clarification / improvement of some ambiguities.

1.     The introduction written by the authors is a really detailed and comprehensive overview of the problems. However, before starting the “Materials and Methods”, more information about the last level of research of other perovskite/oxide-polymer composites should be important to give. This will arouse more keen interest among readers working in related fields. See, some examples (and references therein): 

(BaZrO3):  Savchyn, V. P., Popov, A. I., Aksimentyeva,  et al (2016). Cathodoluminescence characterization of polystyrene-BaZrO3 hybrid composites. Low Temperature Physics42(7), 597-600.

(SrTiO3): Kadhim, A. F., & Hashim, A. (2023). Fabrication and augmented structural optical properties of PS/SiO2/SrTiO3 hybrid nanostructures for optical and photonics applications. Optical and Quantum Electronics55(5), 432.

2.     “Materials and Methods”.  Lines 104-108. What is the crystal structure of BaTiO3 and SiC particles

3.     The two pictures in Fig. 2 are poorly distinguishable and need to be improved.

4.     Fig.3 and corresponding text. There is no comparison with the experiment of other authors.

5.     Table 1. These data need error bars and corresponding discussion in the text, because it seems that there is excessive precision.

6.     Fig.8a. What is the shape of the curves at low frequencies? Same or different?

7.     Has porosity been tested and does this affect the results?

 In general, the manuscript is interesting and can be recommended for publication after constructive reflection on the above comments.

Author Response

Dear Reviewer,

Thank you for the constructive comments.

We addressed the comments you have given us and improved the manuscript accordingly. Our response is attached. 

Sincerely, 

Reviewer 3 Report

This manuscript describes a method of enhancing dielectric properties, thermal conductivity, and mechanical properties of poly(lactic acid)–thermoplastic polyurethane blend composites and SiC/BaTiO3 hybrid filler. Though the study has conducted many experiments, the results are not well discussed and justified. I am not able to recommend acceptance of this manuscript. There are many issues that should be corrected, some of them are pointed out below: 

What is the hypothesis behind using BaTiO3/SiC particles? Please briefly include it in the Abstract and Introduction.

The introduction section needs a good revision. The gap in the research is not very clear. Authors need to demonstrate how their study is filling the research gap in the field.

Fig 1. This should be moved to results discussion and more detailed explanations should be provided with appropriate characterisation, such as FTIR and XPS. For that, those discussions should also be revised to reflect these changes.

"The composites were produced using the melt-extrusion method at a low speed to 138 increase the viscosity" not sure what it means. What was the method of producing the shapes from the melt?

The major limitation of this paper is almost all discussions in the Results have no link to references. There is no justification and validation for the results obtained. It cannot be understood how these results are compared to previous studies. This needs a complete revision. This is unusual that within the first two paragraphs of the paper, it has cited 46 references and probably only 2 in the results part. 

Author Response

(The authors gave the same response as above.)

Reviewer 4 Report

Incorporating inorganic materials in organic polymer materials to play the advantages of two types of materials can obtain significant advantages of composite materials. In this investigation, TPU-PAL was introduced in dielectric inorganic materials to improve their mechanical, thermal conductivity, which has a positive significance for the comprehensive performance of BaTiO3  dielectric materials. However, the manuscript version has the following problems, the authors are advised to consider.

1. FIG. 2 is mentioned at the end of introduction section, however, FIG. 1 is not mentioned before mentioning FIG. 2.

2. In the materials section of "Materials and Methods", MDI was not mentioned.

3. In line 116-117, The solution was washed with DI water until it became neutral.”, How to wash the solution?

4. Line 123-124, how to deal with SiC was not clearly expressed.

5. A schematic diagram,how the modified functional groups to react with the polymers,should be added in Figure 1. In the text, the effects of these modifications on the performances of composite materials should be stated. In Figure 2, the role of each step should be marked.

6. In “the results and discussion” section, it should be paragraphed according to the obvious topics.

7. The method of ATR or KBr pellet used in IR measurement should be indicated. In this investigation, the modification only occurred on surface of materials, so, ATR is more appropriate. The characteristic peaks in the spectra of IR given in Figure 3 were not obvious.For example, in Figure 3a, the characteristic peak of OH at about 3000cm-1 was barely visible, the isocyanate functional groups at 2260 cm-1 originated from the MDI was hardly observed also.

8. Line 184-185, "... OH groups arising from the NaOH". Such expression is inappropriate; In the Figure 3 (a), the difference in spectra between the two samples was not obvious; Figure 3 (b), MDI spectrum should be added and the corresponding characteristic peaks should be marked in Figure 3.

9. In Figure 3, the electron binding energies being present different chemical environments of  N element involved in N1s spectrum were not clearly analyzed. For example, in line 204-207, “Figure 3(d) shows the deconvolution results for MDI-modified BaTiO3 particles. N1s was deconvoluted in the range of 409 and 393 eV binding energies, and the results of deconvolution showed a dominant sharp peak conforming to the amine groups at 399 eV (C=N-C).”. This statement could not observed in the Figure 3d.

10. In Figure 4, the difference of SEM images of BaTiO3 before and after MDI modification was clear. However, the features of difference was hard to be found, especially caused by the modification. In fact, the modification involved processes of NaOH and MDI treatments. However, authors attributed the differences between SEM images of Figures 4b and Figure 4c only to the MDI modification. This conclusion is only valid after excluding the effect of sodium hydroxide.

11. References should be given for the interpretation of the thermal decomposition results for TPU and PLA.

12. In the SEM images showed in Figure 5, the morphology of TPU-PLA is different from its inorganic particle composite material. The cross-section morphology of TPU/PLA indicated the obvious the feature of brittle break. However, the composite materials dis not have the similar brittle fracture of the cross-section morphology. The statements in line 282-287 did not explain clearly why the composite dis not have brittle fracture.In fact, the interface is only a part of the composite material, but more non-interface parts.

13.  The elaborations performed in line 290-307 should be moved to the experimental section rather than the results and discussion section.

14. In line 140, the mixture ratio was TPU/PLA, while in line 144, the ratio is expressed as PLA/TPU. This ratio should be consistent in the text. Otherwise, the relevant study results cannot be analyzed.

15. In line 313-315, 10TPU/PLA has lower tensile strength. Authors attributed this result to the little filled particles. Following this logic, the TPU/PLA without the particles should have lower tensile strength than 10TPU/PLA. But that is not the case (see Figure 6a). Therefore, such an explanation is not reasonable enough. In general, the introduction of solid particle filler in polymer materials will enhance its mechanical strength. In Figure 6 (a), why the tensile strength of the composite materials was less than TPU/PLA.

16. In Figure 6, why is the TPU/PLA not in the normal proportional order ?

17. Line 326-328, authors pointed out As the loading percentage of BaTiO3/SiC particles in the matrix of TPU/PLA increased, the brittle nature decreased, inferring that the composites were more ductile than 60-40TPU-PLA, which might be due to the filler particles incorporation in the TPU-PLA blend matrix.. “However, the elongation at the break point of TPU/PLA composites with MID-functionalized BaTiO3/SiC particles showed a superior ductile property compared with TPU/PLA . How to explain the above seemingly contradictory phenomena.

18.  The conclusion is the judgment of the hypothesis based on the experimental results. For the purposes of this study, based on all experimental results, several explicit judgments should be listed in the conclusions. That is, which factors affect the dielectric properties, thermal conductivity and mechanical properties, do not need to explain. In order to provide reference values for the related researches, it is recommended to give several clear judgmental statements in the conclusion.

Author Response

(The authors gave the same response as above.)

Round 2

Reviewer 3 Report

I believe authors have done the corrections reasonably and explained their points well. The paper can be accepted now.

Author Response

Dear Reviewer,

Thank you for considering our revised version of the manuscript and accepting it for publication. 

Sincerely, 

Author Response

Dear Reviewer,

Thank you for the comments.

However, the comments given to us are not related to our manuscript. We did not study microbes, we studied about the improvement of dielectric constant, thermal conductivity and mechanical property of TPU-PLA blend composites with SiC/BaTiO3 as fillers. 

Thank you!